# Domain Prompt Matters a Lot in Multi-Source Few-Shot Domain Adaptation

## Abstract

Large vision-language models have demonstrated strong performance in multi-source few-shot domain adaptation (MFDA). Advanced methods like CoOp rely on identifying a domain-agnostic prompt, leading to the overlooking of known difference information between domains. However, extracting the domain information requires the model to have good identification ability for domain information. Although training models with domain prompts allows them to capture the specific semantic nuances of a particular domain, using learnable prompts increases the risk of over-fitting on training samples and reduces the effectiveness of domain prompts in capturing target domain features during transfer. To address this challenge, we propose "domain-aware mixup," a method that allows the model to become more sensitive to specific domain information when facing cross-domain mixed feature information. Specifically, we design the prompt structure composed of domain prompt and context prompt to narrow the gap between the specific domain feature and the specific image feature extracted from the cross-domain mix feature. This approach enables us to efficiently train domain prompt terms, enhancing the model's ability to distinguish semantic distinctions between domains. We empirically validate our method on the DomainNet and OfficeHome datasets, observing a performance boost of 5.3%-5.8% over the CLIP model and a 1.1%-1.5% advantage over the domain-agnostic tuning method.

## 1 Introduction

Multi-source domain adaptation (MDA) (Mansour et al., 2012; Duan et al., 2012b; Xu et al., 2018) aims to transfer the task knowledge from multiple fully labeled source domains to an unlabeled target domain. However, with limited labels within each source (She et al., 2020; Cao, 2020), traditional MDA methods may struggle to differentiate between features specific to the target domain and those from the source domain (Yue et al., 2021a). Compared to the traditional MDA method, Contrastive Language-Image Pretraining (CLIP) (Radford et al., 2021a) has gained attention for its impressive performance in few-shot and zero-shot scenarios, which shows potential in tasks with limited samples for this large-scale vision-language model.

However, current prompt learning methods predominantly always pay their emphasis on pursuing a prompt that can universally apply to all domains (Zhou et al., 2022), which ignores the obvious differences in domain information. These domain-agnostic prompt often results in oversight of the inherent disparities between these diverse domains and reduces the transfer ability towards the target domain. Unfortunately, extracting the domain information requires the model to have good identification ability for domain information, which is what domain-agnostic prompt models lack.

Different from finding domain-agnostic prompts, Domain Adaptation Prompt learning (DAPL) (Ge et al., 2022) highlighted the importance of employing domain prompts. DAPL enables the model to have the ability to distinguish between the source domain and target domain by adding source domain-specific tokens to the source domain prompt. By modifying the prompt structure, DAPL achieves promising outcomes in the context of UDA problems. However, in the case of multi-source domains, the extraction of domain information becomes more complex. Using learnable prompts increases the risk of overfitting on training samples, which reduces the ability of domain prompt models to extract common semantic features. Furthermore, the sparsity of training samples amplifies the challenge of domain prompt learning.

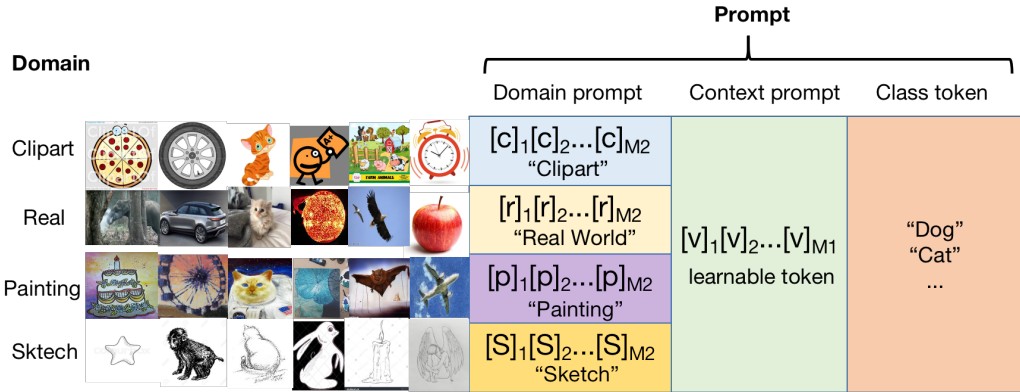

Figure 1: Domain prompt learning: We divide prompts into domain prompts and context prompts. Domain prompts are composed of learnable vectors and domain names, and are not shared between domains. Context prompts are composed of learnable vectors. By training the specific domain prompts to extract the specific information from the mixed feature, the model becomes more sensitive to specific domain information when facing cross-domain mixed feature information.

In this paper, to address the challenge of extracting multiple domain semantic features under the constraint of limited samples, we introduce a new domain prompt learning method. We design the unique domain attributes into each domain prompt and combine them with the context prompt. Importantly, to enhance the effectiveness of the domain prompt in extracting domain-specific features, we have introduced a novel approach method "domain-aware mixup", which enables us to efficiently train domain prompt terms, enhancing the model's ability to distinguish semantic distinctions between different domains. We train the model to become more sensitive to specific domain information when facing cross-domain mixed feature information by narrowing the gap between the specific domain feature and the specific image feature extracted from the cross-domain mix feature.

In summary, our contributions are three-fold: 1) We propose a novel prompt learning method that combines domain prompts with class prompts called multi-source domain prompt learning (MS-DPL). This enables the model to focus on both the class-specific semantic features and the domain semantic features simultaneously, and enhancing the performance of few-shot domain adaptation tasks by opening the difference between different domains, 2) We introduce a cross-domain feature-level mixup method called domain-aware mixup. This strategy, compared to an image-level mixup, is more readily accepted by pre-trained large-scale models and aids in learning domain semantic features. 3) We have carried out comprehensive experiments to validate the effectiveness of our proposed methods. Our method outperforms state-of-the-art domain adaptation methods across multiple benchmark datasets.

## 2 RELATED WORKS

### 2.1 MULTI-SOURCE DOMAIN ADAPTATION

MDA approaches (Sun et al., 2015; Zhao et al., 2019) assume the presence of multiple fully labeled sources and aim to transfer knowledge to an unlabeled target domain. Theoretical analyses (Ben-David et al., 2010; Crammer et al., 2008; Mansour et al., 2008; Hoffman et al., 2018) have been put forth to underpin existing MDA algorithms. Initial MDA techniques often either establish a shared feature space encompassing all domains (Duan et al., 2009; Sun et al., 2011; Duan et al., 2012a;b) or amalgamate pre-learned predictions from source classifiers to yield final predictions using ensemble methods. With the rise of deep neural networks, numerous deep learning-based MDA methods have been introduced, including DCTN (Xu et al., 2018), M3SDA (Peng et al., 2019), MDAN (Zhao et al., 2018), MFSAN (Zhu et al., 2019), and MDDA (Zhao et al., 2020). All these MDA strategies strive to mitigate domain shifts through auxiliary distribution alignment objectives. SImpAl (Venkat et al., 2020) is devised to carry out implicit domain alignment via pseudo-labeling, without introducing

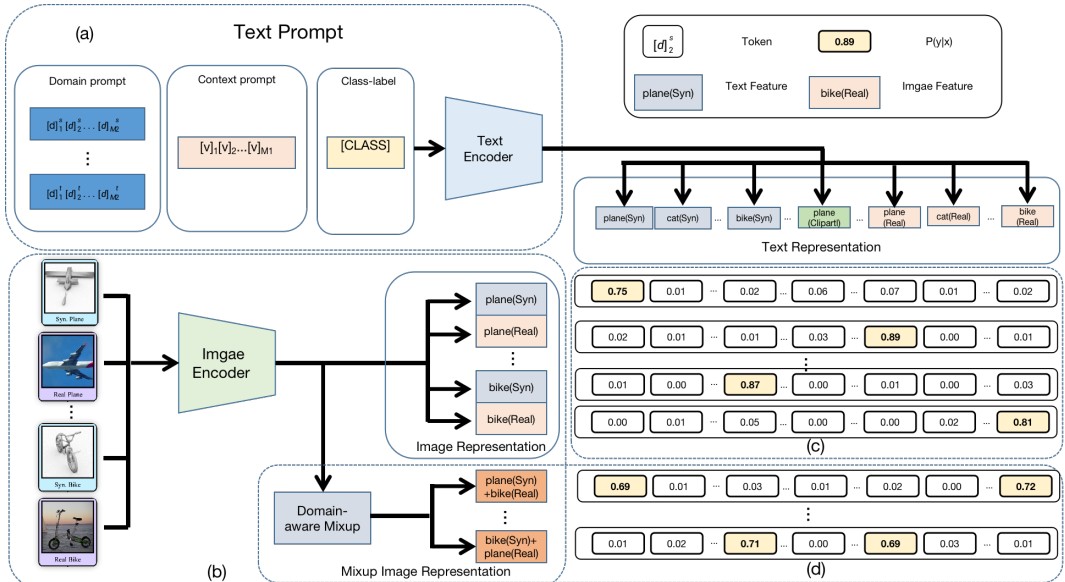

Figure 2: Multi-source Domain Prompt Learning (MSDPL): (a) MSDPL trains the learnable context variables: context variables and domain variables, [CLASS] token and [DOM] token which are combined and encoded by a text encoder. (b) An image encoder encodes images from different domains. (c) Next, cosine similarity between text and image features is computed and the positive pairs (with matched domain and class) are encouraged to align. (d) Computing cosine similarity between domain text feature and mixup image feature. the rare domain pairs are encouraged to align. The classification probability is defined in Eq 6. and a cross-entropy loss is applied between the image feature and the ground truth class to train the networks.

extra training objectives for adaptation. More recently, ProtoMDA (Zhou et al., 2021) proposes the utilization of prototypes for MDA, achieving state-of-the-art performance.

## 2.2 PROMPT LEARNING

Prompt learning, introduced by Petroni et al. (2019), has garnered significant attention within the field of Natural Language Processing (NLP) in recent years (Jiang et al., 2020b; Lester et al., 2021; Li & Liang, 2021; Liu et al., 2023). This methodology involves augmenting input with preparatory instructions, thereby facilitating language model pre-training to enhance downstream task performance. Notably, Petroni et al. (2019) and Poerner et al. (2019) have leveraged manually crafted prompts to bolster language model efficacy. However, the inherent optimality of manually devised prompts can sometimes be suboptimal or even unsuitable, potentially leading to inaccurate guidance.

To extract more precise insights from language models, various approaches have emerged to autonomously explore optimal prompts (Jiang et al., 2020b; Shin et al., 2020; Zhong et al., 2021). Recent developments have extended the concept of prompts to the realm of vision-language models, enabling the acquisition of versatile visual representations (Jia et al., 2021; Radford et al., 2021b). Noteworthy contributions include ALIGN (Jia et al., 2021)and CLIP (Radford et al., 2021a), with CLIP attaining cutting-edge visual representations through language-based supervision across a dataset of 400 million image-text pairs. Furthermore, Zhou et al. (2022) have embraced continuous representations to model prompts, resulting in the automated acquisition of task-relevant prompts, as exemplified by their work named CoOp. In contrast to CoOp, which focuses on formulating domain-agnostic prompts for visual recognition tasks, Ge et al. (2022) introduces a novel domain-aware prompt learning paradigm. By contrasting extensive source domain data with target domain data, the model becomes capable of discerning the disparities between domains.

## 3 METHOD

We study a multi-source few-shot domain adaptation (MFDA) scenario, where a single target domain with sparse labels and multiple source domains with partial labels exist. For the $i$-th source domain, a modestly labeled set $S_i = \left\{ \left( \mathbf{x_i^j}, y_i^j \right) \right\}_{j=1}^{N_i}$ is drawn from the source distribution $p_i (\mathbf{x}, y)$, where $N_i$ represents the count of labeled instances in domain $i$. Within the target domain, $\tau = \left\{ \mathbf{x}_T^j \right\}_{j=1}^{N_T}$ symbolizes the target data subset derived from the target distribution $p_T (\mathbf{x}, y)$ with label observation, where $N_T$ specifies the volume of target instances. Both $N_i$ and $N_T$ are significantly smaller than the total number of samples provided in the training set of the dataset. Our objective is to develop a domain adaptation model capable of accurately predicting target sample labels, while training on $\{S_i\}_{j=1}^{N_i}$ and $\tau$.

### 3.1 PRELIMINARIES

CLIP (Radford et al., 2021a) is trained using paired image-text data in a contrastive manner. Each text input describes a category using the structure "an image of a [CLASS]", where [CLASS] represents the category token. A positive pair, or match, between an image $\mathbf{x}_i$ and its corresponding text $\mathbf{t}_i$, which provides details about the category of $\mathbf{x}_i$. Conversely, a negative pair, or mismatch, occurs when an image $\mathbf{x}_i$ is paired with an unrelated description $\mathbf{t}_j$, where $j \neq i$ is within the mini-batch. The main objective of the training is to increase the cosine similarity for matched pairs while reducing it for mismatched pairs. This contrastive learning objective ensures that both image and text representations are aligned in a unified feature space.

By leveraging these aligned features, the model is capable of making zero-shot predictions. Given $K$ category descriptions, an image $\mathbf{x}$ is classified into the category $\hat{y}$ that exhibits the highest similarity:

$$P(\hat{y} = i \mid \mathbf{x}) = \frac{\exp\left(\langle g\left(\mathbf{t}_i\right), f(\mathbf{x})\rangle / T\right)}{\sum_{k=1}^{K} \exp\left(\langle g\left(\mathbf{t}_k\right), f(\mathbf{x})\rangle / T\right)} \tag{1}$$

$$\hat{y}_i = \arg\max_k P\left(\hat{y}_i = k\right) \tag{2}$$

where $T$ is a user-defined hyperparameter (temperature).

The mentioned input text is a crafted prompt, consisting of a series of distinct tokens. These handcrafted prompts are converted into consistent vectors within the word embedding dimension. Given that such vectors might not be the best representation for distinct categories, there's potential to refine the continuous embedding of these tokens. This continuous characterization $\mathbf{t}_k$ provides a finer depiction of semantic attributes crucial for understanding context variables.

Existing prompt learning methods adopt a domain agnostic style that context is shared across all domains and all categories(Zhou et al., 2022). It follows a unified style:

$$\mathbf{t}_k = [\mathbf{v}]_1 [\mathbf{v}]_2 ... [\mathbf{v}]_{M1} [CLASS]_k \tag{3}$$

where $[\mathbf{v}]_{m1}, m1 \in \{1, 2, 3, ..., M_1\}$ is a vector with the same dimension as the word embedding, and $M_1$ is the number of context tokens applied in the prompt.

### 3.2 DOMAIN PROMPT LEARNING

In the domain of conventional prompt learning, prompts often lack domain distinction. Previous attempts have traditionally aimed to acquire a universally applicable prompt capable of spanning all domains. However, this pursuit has inherent limitations. Large-scale pre-trained vision-language models lack domain awareness during training, which hinders the acquisition of domain-specific tendencies necessary for effective domain adaptation tasks. These models often misinterpret source-domain-specific semantic features as belonging to the target domain, resulting in suboptimal performance for multi-source domain adaptation through prompt learning.

To address the insufficiency of non-discriminative domain prompts in effectively accommodating domain distribution shifts, we propose a paradigm tailored to multi-source domain migration. We

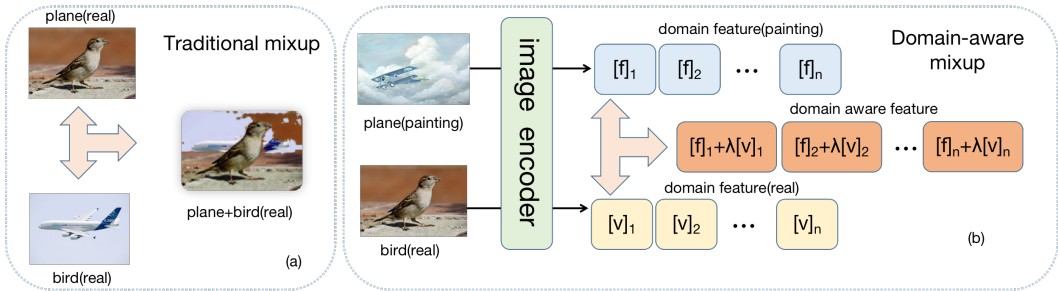

Figure 3: Domain-aware mixup: (a) traditional mixup, which uses two pictures from the same domain with different labels; (b) domain-aware mixup, we combined two features from different domains with different labels through the image encoder.

divide prompts into two segments: domain prompts and context prompts. The former embodies domain knowledge, while the latter transcends domain boundaries. Customized domain prompts are individually tailored for each domain, resulting from the concatenation of learnable parameters with a fraction incorporating unlearnable parameters, as the formulation follows:

$$\mathbf{t}_d = [\mathbf{d}]_1[\mathbf{d}]_2...[\mathbf{d}]_{M2}[DOM]_d \tag{4}$$

where $[\mathbf{d}]_{m2}, m2 \in \{1, 2, 3, ..., M_2\}$ with the same dimension as the word embedding, and $M_2$ is the number of context tokens applied in the prompt.

To emphasize differentiation across domains, non-learnable components are infused with domain-specific expressive terms to enhance distinctiveness and facilitate convergence. Diverse prompts are achieved by constructing a prompt bank that contains various configurations specific to each domain. During model training, expressive terms associated with the respective domain are selectively retrieved from the prompt bank to meticulously craft prompts that meet specific requirements. Context prompts consist of a concatenation of partially learnable parameters harmonized with class labels, providing a comprehensive view across domains.

The designed prompt architecture aims to imbue context prompts with broader communal semantic expressions while including domain prompt components to enhance domain-specific semantic information. In its final form, the prompt configuration crystallizes as follows:

$$\mathbf{t}_k^d = [\mathbf{d}]_1[\mathbf{d}]_2...[\mathbf{d}]_{M2}[DOM]_d[\mathbf{v}]_1[\mathbf{v}]_2...[\mathbf{v}]_{M1}[CLASS]_k \tag{5}$$

The presence of $d * k$ categories arises due to distinct prompts, $\mathbf{t}_k^s$ for the source domain and $\mathbf{t}_k^u$ for the target domain. When presented with a collection of training samples $\{\mathbf{x}_i^s, y_i^s\}_{i=1}^{N_i}$ from the source domain, we have the means to derive the likelihood that a given training sample pertains to the $k$-th category:

$$P(\hat{y}_i^s = k \mid \mathbf{x}_i^s) = \frac{\exp(\langle g(\mathbf{t}_k^s), f(\mathbf{x}_i^s)\rangle / T)}{\sum_{d \in \{s,u\}} \sum_{j=1}^{K} \exp(\langle g(\mathbf{t}_j^d), f(\mathbf{x}_i^s)\rangle / T)} \tag{6}$$

With the probability of the image xi belonging to class $k$, we minimize the standard cross-entropy loss given ground truth label $y$. The loss is computed as follows:

$$L_s = -\frac{1}{N_s} \sum_{i=1}^{N_s} \log P(\hat{y}_i^s = y_i^s). \tag{7}$$

### 3.3 DOMAIN-AWARE MIXUP

In the setting of few-shot learning, learnable prompts exhibit a pronounced propensity for overfitting, making domain differentiation a challenging endeavor. To enhance the model's capacity for domain information assimilation, we utilize cross-domain feature-level mixup.

To optimally extract semantically relevant information from multiple source domains that benefit the target domain, we employ a training scheme that solely integrates the target domain with the source

domain at the feature level. Notably, we discard inter-source domain mixup. This decision aims to prevent the intermingling of source domains from contaminating the representation and learning that are pertinent to the target domain.

Initially, we randomly draw source domain data, denoted as $\mathbf{x}_s^i$, and target domain data, denoted as $\mathbf{y}_t$, from the training samples. Subsequently, we pass these data through the image encoder, resulting in the corresponding image features $\mathbf{f}_s^i$ and $\mathbf{f}_t$. We then amalgamate these features at certain ratios to yield a new feature, which is denoted as $\mathbf{f}_{mixup}^i$. An illustrative overview of our domain-aware mixup is presented in Figure 3.

$$\mathbf{f}_u = (1 - \gamma)\,\mathbf{f}_s^i + \gamma \times \mathbf{f}_t^j \tag{8}$$

where $\mathbf{f}_u$ is the mixed feature and $\gamma$ is the rate of mixup, in this paper we set it as a random number between 0.4 and 0.6.

We consider this new feature as an intermediate state, embodying both source and target domain semantic information. Our aspiration is for the model to possess the capability to discern domain features from such mixed features. We train the prompt of the domain with this mixed-up feature. The specific loss computation is depicted in the subsequent formula:

$$L_u = -\frac{1}{N_u} \sum_{j=1}^{N_u} \left( (1 - \gamma) \log P\left(\hat{y}_j^s = y_j^s \mid \mathbf{t}_k^s\right) + \gamma \log P\left(\hat{y}_j^s = y_j^s \mid \mathbf{t}_k^u\right) \right) \tag{9}$$

where $\mathbf{t}_k^s$ is the specific-domain text feature and the $\gamma$ is the same rate of domain-aware mixup.

In a holistic view, our proposed domain adaptation via domain-aware mixup technique can be seamlessly trained through an end-to-end approach, facilitated by a comprehensive contrastive loss framework.

$$\mathcal{L} = \mathcal{L}_s\left(D^s\right) + \mathcal{L}_u\left(D^u\right) \tag{10}$$

Unlike prevailing domain adaptation approaches that primarily focus on training classifiers for the source domain to capture a conditional probability distribution $P(y \mid \mathbf{x}_s)$, our method operates on a more intricate level. By orchestrating the learning of dual conditional probability distributions, namely $P(y \mid \mathbf{x}_s)$ and $P(y \mid \mathbf{x}_u)$, through the acquisition of distinct sets of prompts $\mathbf{t}_s^k$ and $\mathbf{t}_u^k$ (where $k \in \{1, 2, ..., K\}$), MSDPL transcends the conventional norm. This distinctive attribute empowers our method to nimbly address the challenges posed by both conditional distribution shifts and marginal distribution shifts, thus mitigating the risk of performance degradation that some techniques might encounter (Wang et al., 2020). An illustrative overview of our method framework is presented in Figure 2.

## 4 EXPERIMENT

### 4.1 DATASETS

Following the guideline in Yue et al. (2021a), we put our approach, MSDPL, to the test in a multi-source few-shot scenario on two widely used domain adaptation benchmarks, namely Office-Home (Venkateswara et al., 2017), and DomainNet (Peng et al., 2019). Following the protocols in Yue et al. (2021a), we select the labeled data in each domain, and each domain is alternately treated as the target domain, while the remaining domains within the same dataset serve as source domains. The number of labeled samples we have selected is much smaller compared to the total sample count. Office-Home (Venkateswara et al., 2017), a more challenging dataset, includes four domains (Art, Clipart, Product, Real) distributed over 65 classes. Following the guidelines in, we test settings with 3% and 6% labeled source images per class, leading to an average of 2 to 4 labeled images per class. DomainNet (Peng et al., 2019) is an expansive domain adaptation benchmark. Considering that some domains and classes could be noisy, we adhere to Saito et al. (2019); Yue et al. (2021b) and utilize a subset comprising four domains (Clipart, Painting, Real, Sketch) over 126 classes. For this dataset, we present results from settings with 1-shot and 3-shot source labels.

### 4.2 IMPLEMENTATION DETAILS

We use the pretrained CLIP (Radford et al., 2021a) model based on ResNet-50 (Liu & Tuzel, 2016) as backbones for all baselines. For Office-Home, we fix the parameters in the encoders and the

Table 1: Adaptation accuracy (%) with 3% and 6% labeled samples per class on Office-Home dataset

| | | Office-Home | | | | | | | | | |
| | | 3% | | | | | 6% | | | | |
| | method | Ar,Pr,Rw ->Cl | Cl,Pr,Rw ->Ar | Cl,Ar,Rw ->Pr | Cl,Ar,Pr ->Rw | Avg | Ar,Pr,Rw ->Cl | Cl,Pr,Rw ->Ar | Cl,Ar,Rw ->Pr | Cl,Ar,Pr ->Rw | Avg |
|---|---|---|---|---|---|---|---|---|---|---|---|
| Source-Only | single-best | 29.0 | 41.2 | 52.3 | 43.1 | 41.4 | 36.0 | 49.9 | 61.8 | 54.6 | 50.6 |
| | ComBined | 42.2 | 55.3 | 63.6 | 64.1 | 56.3 | 45.3 | 60.4 | 70.5 | 70.9 | 61.8 |
| Single-best | CDAN | 27.0 | 38.7 | 44.9 | 40.3 | 37.7 | 40.1 | 54.9 | 63.6 | 59.3 | 54.5 |
| | MME | 29.0 | 39.3 | 52.0 | 44.9 | 41.3 | 37.3 | 54.9 | 66.8 | 61.3 | 55.1 |
| | MDDIA | 29.5 | 47.1 | 56.4 | 51.0 | 46.0 | 37.1 | 58.2 | 68.4 | 64.5 | 57.1 |
| | CDS | 37.8 | 51.6 | 53.8 | 51.0 | 48.6 | 45.3 | 63.7 | 68.6 | 65.2 | 60.7 |
| | PCS | 52.5 | 66.0 | 75.6 | 73.9 | 67.0 | 54.7 | 67.0 | 76.6 | 75.2 | 68.4 |
| Source-conbined | CDAN | 52.3 | 52.3 | 64.5 | 63.2 | 55.7 | 51.1 | 67.0 | 74.2 | 73.3 | 66.4 |
| | MME | 34.6 | 55.4 | 67.4 | 64.5 | 57.5 | 46.0 | 67.1 | 75.5 | 75.7 | 66.1 |
| | MDDIA | 63.4 | 66.9 | 72.3 | 75.3 | 67.5 | 57.3 | 67.2 | 79.0 | 74.4 | 66.5 |
| | CDS | 54.9 | 66.2 | 71.6 | 73.4 | 66.5 | 54.9 | 67.5 | 76.1 | 77.5 | 69.0 |
| | PCS | 49.4 | 67.0 | 75.0 | 76.3 | 66.9 | 50.4 | 67.0 | 77.8 | 79.4 | 68.7 |
| Multi-source | SImpAI | 46.8 | 56.7 | 65.1 | 66.6 | 58.8 | 49.3 | 62.1 | 71.7 | 73.0 | 64.1 |
| | MFSAN | 36.9 | 46.6 | 58.9 | 55.6 | 50.3 | 44.5 | 53.7 | 65.4 | 64.2 | 57.0 |
| | PMDA | 50.8 | 56.8 | 64.2 | 66.7 | 59.7 | 54.4 | 65.8 | 70.4 | 71.8 | 65.6 |
| | MSFAN | 55.6 | 60.4 | 70.6 | 76.6 | 69.1 | 56.3 | 68.7 | 79.3 | 79.1 | 70.9 |
| Large-model | CLIP | 51.9 | 71.6 | 81.5 | 82.5 | 71.8 | \ | \ | \ | \ | \ |
| | CoOP | 56.9±0.1 | 74.0±0.1 | 85.7±0.2 | 84.4±0.2 | 75.2±0.1 | 58.3±0.1 | 74.6±0.1 | 86.9±0.1 | 85.6±0.2 | 76.3±0.1 |
| | MSDPL(Ours) | **57.6±0.1** | **75.2±0.2** | **86.7±0.2** | **85.6±0.1** | **76.3±0.1** | **60.0±0.1** | **76.5±0.1** | **88.2±0.1** | **86.4±0.2** | **77.8±0.1** |

prompt is trained with the mini-batch SGD optimizer for 12 epochs, where the batch size is set to be 32. The initial learning rate is set to 0.005 and decayed with a cosine annealing rule (Kim et al., 2020). For domainnet, the encoder parameters are kept fixed, while the prompt is trained to utilize a mini-batch SGD optimizer for 20 epochs with a batch size of 32. We start with a learning rate of 0.005 and apply a cosine annealing rule for its decay. As for the hyperparameters, the length of context tokens $M_1$ and domain tokens $M_2$ are both set to 4. Our context vectors are randomly initialized using a zero-mean Gaussian distribution with a standard deviation of 0.02. The rate of the mixup is both set to 0.5 for Office-Home and Domainnet and we take the average of the three results as the accuracy of the results.

## 4.3 RESULTS ON MFDA

### 4.3.1 BASELINE

In our study, we evaluate the multi-source domain prompt learning method alongside various other techniques to conduct a comparative analysis. Firstly, we consider the "source-only" strategies, where models are trained on labeled data from source domains and directly tested on the target domain. Secondly, we examine single-source domain adaptation methods, which approach multi-source DA by considering single-source DA. The models included in this category are CDAN (Long et al., 2018), MDDIA (Jiang et al., 2020a), MME (Saito et al., 2019), CDS (Kim et al., 2020), and PCS (Yue et al., 2021b). Notably, CDS (Kim et al., 2020) and PCS (Yue et al., 2021b) are specifically developed for single-source few-shot DA (FSDA). Furthermore, we explore multi-source DA approaches, which are designed for MDA and assume the presence of multiple fully-labeled sources. The models considered in this group are MFSAN (Zhu et al., 2019), SImpAl (Venkat et al., 2020), and ProtoMDA (Zhou et al., 2021). It is worth mentioning that SImpAl (Venkat et al., 2020) and ProtoMDA (Zhou et al., 2021) are considered to be cutting-edge, with ProtoMDA utilizing prototypes for MDA. Finally, we investigate approaches that employ large vision-language models, including CLIP (Radford et al., 2021a), CoOp (Zhou et al., 2022), and our method called Multi-source Domain Prompt Learning(MSDPL). We perform a reevaluation of all benchmark methods within the new multi-source few-shot domain adaptation (MFDA) setting and compare them with our proposed method.

A comprehensive series of experiments was conducted across the Office-Home and DomainNet datasets. The results of these experiments are detailed in Tables 1 and 2. Upon analyzing these results, several key insights were discerned:

(1) In the context of single-best, the performance of source-only noticeably exceeds that of certain UDA approaches in multiple conditions. A parallel trend is evident in the source-combined scenario, illustrated by a score of 40.1% versus 31.3% in the Office-Home dataset under similar conditions.

(2) In the MFDA framework, the simplistic strategy of merging multiple sources and executing single-source DA can inadvertently degrade performance compared to sticking to an individual do-

Table 2: Adaptation accuracy (%) comparison with 1 and 3 labeled samples per class on DomainNet.

| | | DomainNet | | | | | | | | | |
| | | 1-shot | | | | | 3-shot | | | | |
| | method | P,R,S ->C | C,R,S ->P | C,P,S ->R | C,P,R ->S | Avg | P,R,S ->C | C,R,S ->P | C,P,S ->R | C,P,R ->S | Avg |
|---|---|---|---|---|---|---|---|---|---|---|---|
| Source-Only | single-best | 18.4 | 30.6 | 28.9 | 16.7 | 23.7 | 30.2 | 44.2 | 49.8 | 24.2 | 34.4 |
| | ComBined | 30.8 | 49.4 | 43.3 | 36.9 | 40.1 | 45.3 | 57.4 | 64.7 | 42.6 | 50 |
| Single-best | CDAN | 16.0 | 25.7 | 19.5 | 12.9 | 18.5 | 30.0 | 40.1 | 40.8 | 17.1 | 29.3 |
| | MME | 16.0 | 29.2 | 26.0 | 13.4 | 21.2 | 25.1 | 46.5 | 50.0 | 20.1 | 32.6 |
| | MDDIA | 18.0 | 30.6 | 27.4 | 15.9 | 23.0 | 41.4 | 50.7 | 52.9 | 23.1 | 38.2 |
| | CDS | 16.7 | 24.4 | 15.9 | 13.4 | 17.6 | 35.0 | 43.8 | 36.8 | 31.1 | 32.9 |
| | PCS | 39.0 | 51.7 | 38.8 | 39.8 | 42.3 | 45.2 | 59.1 | 66.6 | 41.9 | 51.0 |
| Source-conbined | CDAN | 25.7 | 33.0 | 40.0 | 26.4 | 31.3 | 47.8 | 54.1 | 65.6 | 49.1 | 49.6 |
| | MME | 20.0 | 45.3 | 52.5 | 13.0 | 32.7 | 44.2 | 62.7 | 73.9 | 51.8 | 53.1 |
| | MDDIA | 44.0 | 46.4 | 49.6 | 37.1 | 44.3 | 56.3 | 59.3 | 70.3 | 51.3 | 56.3 |
| | CDS | 42.2 | 53.3 | 55.4 | 38.5 | 47.4 | 50.2 | 61.5 | 71.8 | 47.3 | 55.6 |
| | PCS | 36.2 | 53.0 | 56.4 | 32.8 | 44.6 | 45.6 | 61.2 | 74.3 | 41.3 | 53.4 |
| Multi-source | SImpAI | 48.0 | 40.3 | 45.7 | 35.3 | 42.3 | 51.5 | 47.4 | 68.8 | 45.3 | 51.1 |
| | MFSAN | 41.6 | 33.5 | 38.8 | 29.6 | 35.9 | 43.5 | 42.3 | 63.2 | 41.1 | 45.2 |
| | PMDA | 49.3 | 42.2 | 45.0 | 34.8 | 42.8 | 52.2 | 52.5 | 71.3 | 47.6 | 53.3 |
| | MSFAN | 57.3 | 68.7 | 64.8 | 45.2 | 59.0 | 57.8 | 65.5 | 75.8 | 53.6 | 62.3 |
| Large-model | CLIP | 54.7 | 55.4 | 77.1 | 49.2 | 59.1 | \ | \ | \ | \ | \ |
| | CoOP | 59.3±0.1 | 59.6±0.1 | 79.4±0.1 | 52.6±0.2 | 62.7±0.1 | 60.1±0.2 | 60.4±0.1 | 80.0±0.1 | 53.4±0.1 | 63.4±0.1 |
| | MSDPL(Ours) | **59.8±0.2** | **61.0±0.1** | **80.2±0.1** | **53.7±0.3** | **63.7±0.1** | **61.4±0.2** | **60.7±0.1** | **81.1±0.1** | **54.4±0.1** | **64.5±0.1** |

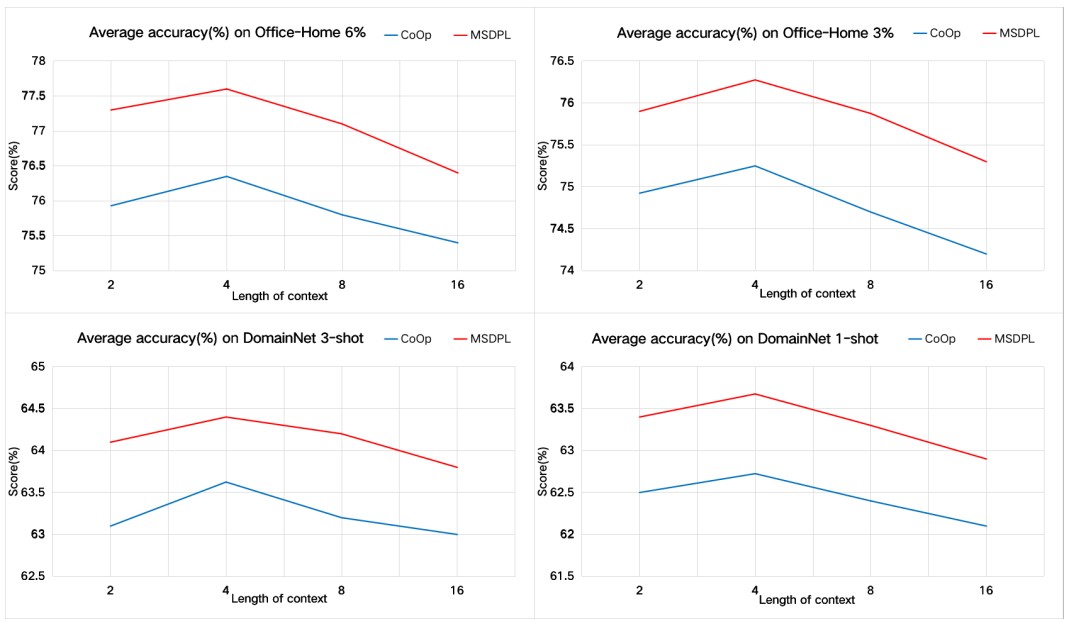

Figure 4: Investigations on context length of MSDPL(Ours) and CoOp.

main. As evidence, a score of 42.3% is registered versus 44.6% in the 1-shot per class category on DomainNet for PCS, a method explicitly tailored for single-source few-shot DA.

(3) Within the MFDA setup, traditional MDA techniques yield subpar results when juxtaposed against single-source DA strategies, a case in point being 65.6% versus 68.7% in the Office-Home dataset when 6% of the labels per class are considered.

(4) Within methodologies that employ large models, the unmodified CLIP model achieves scores of 59.1% and 71.8% on the DomainNet and Office-Home datasets, respectively. In contrast, our proposed approach, which focuses on learning domain and context prompts, achieves a significant improvement in accuracy, specifically 5.3% and 5.8% respectively. Additionally, when compared to the domain-agnostic learning method CoOp (Zhou et al., 2022), our method shows a performance improvement of 1.1% and 1.25% in accuracy respectively.

## 4.4 ABLATION STUDY AND ANALYSIS

We embark on an extensive exploration of the individual components comprising the MSDPL framework within the context of Office-Home. The findings presented in Table 5 unequivocally establish

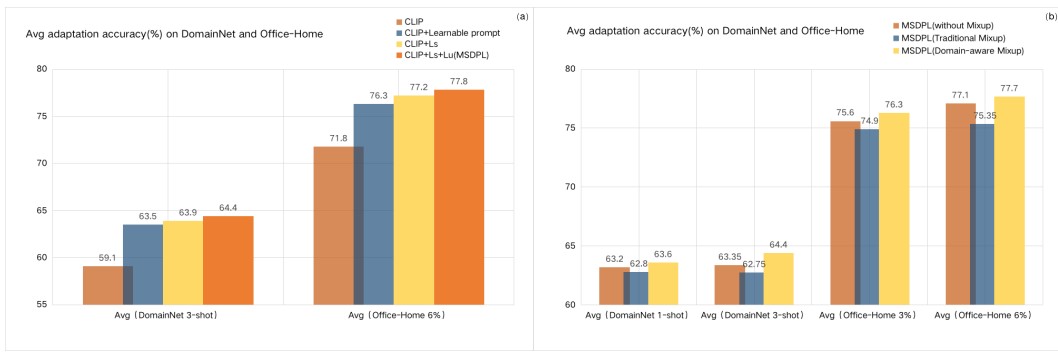

Figure 5: (a) Performance contribution of each part in MSDPL framework on Office-Home and DomainNet. (b)Compare to the traditional mixup, domain-aware mixup enables domain prompt to train more efficiently with few-shot samples.

that the inclusion of each component makes a substantial positive contribution to the ultimate MFDA performance, all while maintaining accuracy at its zenith, and the adoption of domain prompt yields a discernible uptick in model accuracy. Notably, the incorporation of the domain-aware mixup strategy propels accuracy even further beyond the bounds achieved without its deployment. To substantiate our model's acumen in discerning inter-domain disparities, we undertake an experiment whereby target-domain prompts are applied to test other domains. The results underscore that the model excels only when domain cues correspond precisely on a one-to-one basis. This empirical evidence accentuates the sensitivity of our model to domain variations.

Furthermore, to substantiate the merits of our proposed domain-aware mixup, we conducted a comparative analysis against the traditional mixup (Zhang et al., 2017). As demonstrated in Figure 5, the conventional mixup method exhibits enhanced generalization effects for a limited set of domains; however, it yields detrimental consequences in more challenging domains. In contrast, domain-aware mixup showcases significant improvements across all domains. This serves as evidence that our domain-aware mixup technique excels in training domain prompts, enabling the model to adeptly discern disparities between different domains. We also compare the different length of the context. Depicted in Figure 4, the learnable context of size 4 demonstrated significant improvements compared to the others.

## 5 CONCLUSION AND DISCUSSION

In this paper , we investigates Multi-source Few-shot Domain Adaptation(MSFDA), which is a practical and challenging task where each source and target domain has only a small fraction of labeled samples. In limited data scenarios, a novel prompt learning methodology is employed, which utilizes domain prompts to enhance large-scale vision-language models. This augmentation enables the model to effectively identify domain disparities between source and target domains. Additionally, the paper introduces the innovative concept of "domain-aware mixup", which is distinct from conventional mixup approaches that often yield adverse consequences when applied across domains. The novel mixup method significantly aids in the learning of domain prompts, allowing for the acquisition of unique semantic information for each domain.

Our approach demonstrates significantly positive outcomes, validating its effectiveness across a wide range of scenarios. Notably, further investigations reveal a compelling revelation. Traditional learnable prompts tend to have decreased transfer capabilities in domain adaptation tasks when the number of source domain samples is significantly larger than the target domain. In contrast, our method adeptly addresses this challenge and enables proficient domain adaptation despite the disparities in sample sizes between source and target domains.

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

## A  APPENDIX

### A.1  DATASET DETAILS

- **DomainNet** is a versatile benchmark used for evaluating domain adaptation and transfer learning methods. It comprises six diverse domains: Clipart, Infographics, Painting, Quickdraw, Real, and Sketch. Each domain contains the object of 345 categories in different environments. These domains showcase different art styles, data sources, and visual content, making DomainNet an ideal resource for studying cross-domain challenges.

- **Office-Home** is a large-scale benchmark for visual cross-domain recognition. It collects a total of 15,500 images from four distinct domains: Art (Ar), Clip Art (Cl), Product (Pr), and Real World (Rw). Besides, each domain contains the objects of 65 categories in the office and home environments.

### A.2  ADDITIONAL EXPERIMENTAL RESULTS

In this section, we present supplementary experimental material to comprehensively showcase and elucidate the effectiveness of our proposed method (MSDPL). To facilitate a better understanding, we provide the meanings of the symbols utilized in the main formulas through Table 3, presented in a tabular format.

In the main context of this paper, we showcase the performance of various losses within MSDPL on the Office-Home dataset. In order to substantiate the necessity of each loss component, we conduct additional experiments of MSDPL on the DomainNet dataset.

### A.3  HYPERPARAMETER SENSITIVITY ANALYSIS

We conducted an analysis of the hyperparameters in our model, including the size of the learnable context and the mixup ratio $\gamma$. As depicted in Figure 6,7, we observed that the model performs optimally when the size of the learnable context is set to 4. We attribute this observation to the fact

Table 3: Description of key symbols in this paper

| Symbol | Description |
|---|---|
| $M1$ | is the number of the context prompt |
| $M2$ | is the number of the domain prompt |
| $\mathbf{t}_d$ | is the domain prompt of the $d$ domain |
| $\mathbf{t}_k^s$ | is the text feature of the $s$ source domain and the $s$ class |
| $\mathbf{t}_k^u$ | is the text feature of the target domain and the $s$ class |
| $N_s$ | is the number of training simple |
| $N_u$ | is the number of mixup simple |
| $\gamma$ | is the rate of mixup |
| $\mathbf{f}_s^i$ | is the image feature of the $s$ source domain and the $i$ class |
| $\mathbf{f}_t^j$ | is the image feature of the target domain and the $j$ class |
| $D^s$ | is the training data |
| $D^u$ | is the mixup data |
| $L_s$ | is the loss of domain prompt learning |
| $L_u$ | is the loss of domain-aware mixup |
| $L$ | is the loss of model training |

that in few-shot learning scenarios, excessive model flexibility can hinder convergence and effective learning. Consequently, the learnable context of size 4 demonstrated significant improvements compared to the larger size of 16.

Regarding the mixup ratio, we found that the optimal value depends on the strength of the target domain. For instance, in domains with weaker representation, such as "sketch," the model achieved its best results with a mixup ratio skewed towards 0.6. Conversely, in domains with stronger representation, like "real," the model performed better with a mixup ratio leaning towards 0.4. Taking both cases into account, an overall balanced mixup ratio of 0.5 resulted in the most stable performance. This observation may be attributed to the fact that stronger domains contain more domain-specific information, while weaker domains possess relatively less. Hence, maintaining a balanced mixup ratio facilitates the model's ability to learn distinctive domain-specific features effectively.

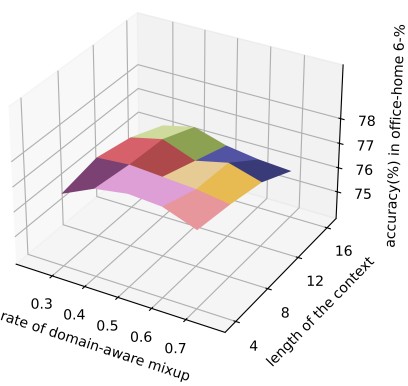

Figure 6: Hyperparameter Sensitivity Analysis on DomainNet

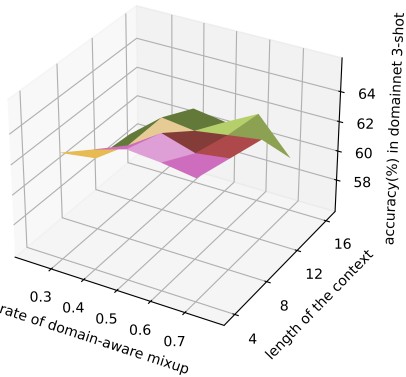

Figure 7: Hyperparameter Sensitivity Analysis on Office-Home

