# OpenReview forum: "Domain Prompt Matters a Lot in Multi-Source Few-Shot Domain Adaptation"
_ICLR.cc/2024/Conference — Submitted to ICLR 2024_

### Official Review · Reviewer_7xZk · 2023-10-26

**Soundness:** 2 fair
**Presentation:** 1 poor
**Contribution:** 1 poor
**Rating:** 1
**Confidence:** 5

**Summary:**

This paper addresses the problem of multi-source few-shot domain adaptation (MFDA). MFDA involves utilizing only a small amount of labeled data from multiple source domains and a few unlabeled data from a target domain. The goal is to transfer knowledge from the source domains to the target domain. The authors propose using the CLIP model and designing two separate pairs of prompts (a domain prompt and a context prompt) to better capture the distinct knowledge of the source domains. Additionally, a feature mixup mechanism is introduced to enhance the sensitivity to domain-specific information. Experiments were conducted on the DomainNet and OfficeHome datasets to demonstrate the superiority of this approach over CLIP and CoOP.

**Strengths:**

- The paper is addressing the challenging setting of MFDA.
- Prompting techniques are utilized to better learn the domain knowledge via the foundation model.
- Superior performance is achieved compared with CLIP and CoOP.

**Weaknesses:**

Many significant problems are found in the current format of the paper that prevents the understanding of the concept. The problem includes but is not limited to confusing writing, inconsistency of notations, expressions of novelty, experimental presentation etc. It is highly recommended that the authors to re-write the paper, re-organize the content and better polish the text for the reader to better understand.

The major problems:
* Novelty is limited:
    - The proposed method as in Sec. 3.2 is very similar to DAPL but extended to multi-source scenarios. The only difference is to introduce an additional [DOM]. Note, that DAPL has not been peer-reviewed.
    - The motivation for domain-aware mixup is confusing. I cannot be convinced and do not understand in the current writing, how it can enforce to learn domain-specific knowledge. The corresponding literature regarding mixup in the feature space is also not referenced and discussed (e.g. [1]).
    - The description for deriving the domain-aware mixup is confusing. I assume the authors are trying to develop a method so that the learned prompt shares the knowledge between source and target domains (depending on Eq. 9)?

* Writing:
    - In the first sentence of Abstract: “large vision-language models … strong performance in MFDA”. There is no such reference applying large VL models in MFDA. In fact, MFDA is a rarely studied problem.
    - The description of the problem setting (MFDA) should be clearly explained at the beginning (abstract or introduction) so that the reader can refer better to the limitations of prior works.
    - Paragraphs 1 & 2 in the introduction: the connection is missing, and ‘prompt learning’ suddenly jumps in, making the concept broken.
    - Fig. 1 is not referred to in the paper.
    - Related work: after describing the related prior works of each field, it's suggested to write a couple of sentences to distinguish between them to show the novelty of the proposed method.
    - The description of the MFDA setting is very confusing in the first paragraph of the Method Section: “single target domain with \textbf{sparse} labels”, “…target distribution p_T(x, y) with label observation…” is mentioned, but the notation for target domain \tau is unlabeled. In the original MFDA paper (Yue et al., 2021a), the target data is unlabeled. What about the unlabeled data in source domains? Are they used during training (as in (Yue et al., 2021a))? It is very confusing that the problem setting description defers significantly as in (Yue et al., 2021a).
    - There is significant text overlapping with DAPL in the preliminary sections of both papers (only with some rewording..). It should be strictly prohibited.
    - What is [DOM] in Eq. 4? I assume it is a domain ID? And I assume [DOM] is the non-learnable component near the description of Eq. 4?
    - Notation: what is subscript d in Eq. 4 and superscript d in Eq. 5? They are not explained in the text. I assume they are the domain IDs?
    - What does it mean by ‘d*k categories’ as in the sentence after Eq. 5?
    - Eq. 6 is very confusing. For the outer summation on d \in {s, u}, what is the purpose of computing the similarity between the target domain prompt and source image features? How does the learning on unlabeled target data is realized?
    - What is inter-source domain mixup? In the current format of writing, I don’t understand why maintaining it will harm the representation learning on the target domain. The motivation is weak.
    - In the second paragraph on page 6, the notation of target domain data y_t is different from Section 3.
    - In Fig. 3, letters v and f are used to represent the features of “painting” and “real”. But v is used to represent text prompts as in Eq. 3
    - The feature mix-up formulation in Fig. 3 is different than Eq. 8. One uses \gamma and another one uses \lambda? and the weighting is different?
    - It is really confusing that the letter “t” is used to refer to text and target domain.
    - What are D^s and D^u in Eq. 10? They are never defined. I assume they are source and target domains, which is inconsistent with what is described in the problem setting. The problem setting is borrowed from (Yue et al., 2021a). But Eq. 10 is copied from DAPL paper. Please keep everything consistent throughout the paper. Also, Eq. 9 requires source data as well, why only D^u is passed to L_u as in Eq. 10?
    - The notations for loss functions in Eq. 7, 9, and 10 should be consistent.
    - Table 5 in the last sentence of Page 8 should be Figure 5.
    - The experimental setting/comparison is very confusing. What is “single best”, which can be both setting and method as in Table 1&2? What is source combined? Which rows in Tables 1&2 refer to the MFDA? How come the “Large model” in Table 1&2 can be the setting, it should be the model architecture.
    - For Figure 6&7, they are hard to see the differences. It is suggested to use a table to report the numbers.

[1] Adversarial Domain Adaptation with Domain Mixup. AAAI 2020.

**Questions:**

- In Eq. 1, what do g() and f() represent? I assume they are text and image encoders as described in DAPL.
- Missing information: “Diverse prompts are achieved by constructing a prompt bank that contains various configurations specific to each domain.” How does the prompt bank is constructed and what are the specific config for each domain?
- What is the relation between f^i_{mixup} and f_{u}? It seems they are the same. But neither f^i_{mixup} or f_{u} is used in the subsequent text?

---

> ### Author Response · Authors · 2023-11-22
>
> I would like to express my gratitude for your thorough and detailed review of my manuscript. Your careful examination has been instrumental in identifying various issues within my paper. Firstly, in response to your feedback on the writing aspects of the paper, we have systematically reviewed and addressed each of the issues you raised, incorporating necessary modifications.
>
> Regarding the novelty concerns raised, I wish to highlight that our primary objective is to introduce a novel training paradigm. By incorporating a domain-aware training mechanism, we aim to provide the model with explicit domain label information for images already present in the domain adaptation dataset. This targeted approach enhances the model's accuracy in prompt learning. Additionally, we propose the concept of "domain-aware mixup" to further improve the model's ability to learn from domain label information in a more refined manner. Through training the model to classify domain information in images containing cross-domain cues, we enable the model to acquire more suitable domain prompts.
>
> Q1.response:
> We appreciate your observation regarding the clarification of functions g() and f() in Equation (1) as the text encoder and image encoder, respectively. In response to this, we have incorporated additional descriptions in the manuscript to explicitly define these functions and enhance reader understanding.
> Q2.response:
> Specifically, our objective is to incorporate x sets of distinct domain prompts in the model, where x corresponds to the number of existing domains. Each domain set is designed to contain a number of prompts equivalent to the number of categories in the image classification task. The intention is to utilize the domain labels during classification to select the corresponding set of domain prompts, facilitating a more targeted and context-aware classification process.
> This approach enables us to align the domain-specific prompts with the domain labels, contributing to more effective and nuanced classification outcomes.
> Q3.response:
> We acknowledge the writing error in our previous statement, and we appreciate your astute observation. We have rectified this error and supplemented additional descriptions to accurately convey that their essence is indeed the same.
>
>
> We hope this clarification effectively conveys the motivation behind our paper. Your rich and detailed feedback has been sincerely appreciated, and we are committed to incorporating these valuable insights into the refinement of our manuscript.
>
> Thank you for your thoughtful evaluation.Your insightful suggestions are invaluable, and we are dedicated to ensuring the precision and clarity of our manuscript.

---

> ### Comment · Reviewer_7xZk · 2023-12-05
> **Comments by Reviewer 7xZk**
>
> To AC:
>
> The authors did not address my questions. Regarding the novelty concern, the response is more likely to rephrase some of the text. On the other hand, the authors did not update the paper draft, therefore, the writing issues have not been addressed. The paper in its current form requires significant modification. I maintain my rating as reject.

---

### Official Review · Reviewer_aew1 · 2023-10-30

**Soundness:** 3 good
**Presentation:** 2 fair
**Contribution:** 2 fair
**Rating:** 6
**Confidence:** 4

**Summary:**

This paper addresses the challenges in multi-source few-shot domain adaptation (MFDA) using large vision-language models. Current methods, like CoOp, utilize a domain-agnostic prompt, which often neglects the differential information between domains. The authors suggest that although training models with domain-specific prompts can help capture the unique semantic nuances of a domain, it also increases the risk of overfitting. To overcome these challenges, the authors introduce "domain-aware mixup," which allows the model to become more attuned to domain-specific information during cross-domain feature mixing. They empirically validated their method on DomainNet and OfficeHome datasets and reported performance improvements over existing models.

**Strengths:**

- The paper addresses a crucial problem in the domain of few-shot domain adaptation.

- The introduction of "domain-aware mixup" is a novel approach to handle the challenges of domain adaptation.

- Empirical validation on standard datasets provides evidence of the proposed method's effectiveness.

**Weaknesses:**

- The paper could benefit from more ablation studies analyzing the effects of different prompts.

- The difference between the proposed method and existing works isn't clearly demarcated.

- Implementation details and settings for the backbone model are missing.

**Questions:**

- Can the authors provide more ablation experiments on the effects of the three types of prompts?

- What exactly is the differentiation between the current work and previous approaches in terms of formulae and figures presented in the paper?

- Would it be possible to test the method on more downstream tasks to evaluate its broader applicability?

---

> ### Author Response · Authors · 2023-11-22
>
> I extend my appreciation for your valuable feedback on my manuscript. We are committed to diligently addressing your suggestions to enhance the overall quality of our paper.
> Q1.response:
> In accordance with your request, we have incorporated additional ablation experiments, providing a more comprehensive comparison with existing works. Furthermore, we have elucidated our improvements through explicit formulation and numerical analysis.
> Q2.response
> In comparison to prior works, our innovation primarily lies in the introduction of [DOM] and the novel training approach. To facilitate a clearer understanding of the innovation behind our method, we have incorporated additional mathematical descriptions. These additions aim to provide readers with a more detailed and comprehensive insight into the unique aspects of our approach.
> Q3.response:
> Thank you for your valuable suggestion. As of the current state of our research, we have not yet explored a broader range of downstream tasks. We appreciate your insight and plan to address this by conducting additional experiments in future research to evaluate the performance across a more extensive set of downstream tasks.
>
> Your insightful comments have been instrumental in guiding our revisions, and we are dedicated to ensuring that the modifications align with the high standards expected in the field.
>
> Thank you for your thoughtful evaluation.

---

### Official Review · Reviewer_4Xat · 2023-10-31

**Soundness:** 2 fair
**Presentation:** 2 fair
**Contribution:** 2 fair
**Rating:** 3
**Confidence:** 5

**Summary:**

The paper presents a domain prompt learning approach designed to extract domain-specific information and tackle domain adaptation within a few-shot setting. Instead of using domain-agnostic prompts shared across all domains, the proposed method distinctively categorizes prompts into two types: domain-specific prompts and context prompts. While domain prompts are specific to their respective domains, context prompts are shared across multiple domains. To further enhance the differentiation of domain-specific information across various domains, the authors introduce a domain-aware mixup technique. The effectiveness of the proposed method has been rigorously validated on several benchmark datasets.

**Strengths:**

* The paper introduces a prompt-learning-based method to address domain adaptation in vision-language models.

* The author empirically validates the method on multiple benchmark datasets.

**Weaknesses:**

* Prepending and learning domain-specific or instance-specific prompt tokens to handle distribution shifts is not a novel idea. For example, to improve generalization to OOD data, the prompt vectors are conditioned on image inputs at test time [1, 2, 3]. Additionally, the idea of dividing prompts into domain-specific and domain-agnostic parts is introduced in [4]. Specifically, [4] heuristically partition the prompts into domain-specific and domain-shareable components. During adaptation, a manually crafted regularization term is employed to preserve the domain-shareable part while allowing the domain-specific component updates.

[1]Conditional prompt learning for vision-language models. CVPR 2022

[2]Domain Prompt Learning for Efficiently Adapting CLIP to Unseen Domains. 2022

[3] Test-time prompt tuning for zero-shot generalization in vision-language models. NeurIPS 2022

[4]Decorate the newcomers: Visual domain prompt for continual test time adaptation. AAAI 2023

* The paper falls short in providing a thorough review of related work. The previous works on domain adaptation with prompt learning between 2022 and 2023 are missed. Several works are mentioned above, and thus I suggest the author undertake a more exhaustive literature review and highlight the differences between the proposed method and those previous works.

* The presentation of the problem setting and motivation requires further clarification. While the paper puts forth a solution for few-shot domain adaptation, it doesn't clearly delineate the inherent challenges of the few-shot setting. Readers might wonder: How do limited labels influence the model's training? Specifically, does this scarcity impact the modeling of multi-source domains or the transfer of knowledge from the source to the target domain?

* The claims in the paper lack evidence. The author mentioned the extraction of domain information becomes more complex in multi-domain settings and using learnable prompts increases the risk of overfitting on training samples. However, those claims are neither supported with references nor empirical results.

* The writing of the introduction may pose challenges for readers who are new to the subject because the author mentions many terms but without adequate explanation. For example, what are context prompts? What are the differences between the domain and the semantics? What are context variables and domain variables? What is cross-domain mixed feature information?

**Questions:**

Please see the section on weaknesses

---

> ### Author Response · Authors · 2023-11-22
>
> I express my gratitude for your insightful comments on my manuscript.
> Q1.response:
> Regarding your suggestion for a comprehensive review of related work, we have diligently examined the references you highlighted and incorporated them into the relevant research section of our paper. In response to your concern about the potential overfitting issue arising from the use of learnable prompts, our investigation was primarily grounded in further experiments inspired by the observations presented in CoCoOp [1] and [2]. The associated problem is briefly addressed in the cited reference, and we acknowledge the need for clarification in our manuscript. We commit to refining this aspect to enhance reader comprehension.
> Q2.response:
> We sincerely apologize for any confusion caused by the lack of clarity in our background introduction. In response to this concern, we have made revisions to enhance clarity, aiming to provide readers with a more accessible understanding of the presented material.
>
> I appreciate the depth of your feedback and the valuable insights you have provided.
>
> [1]Zhou K, Yang J, Loy C C, et al. Conditional prompt learning for vision-language models[C]//Proceedings of the IEEE/CVF Conference on Computer Vision and Pattern Recognition. 2022: 16816-16825.
>
>
> [2] Ma C, Liu Y, Deng J, et al. Understanding and mitigating overfitting in prompt tuning for vision-language models[J]. IEEE Transactions on Circuits and Systems for Video Technology, 2023.

---

> > ### Comment · Reviewer_4Xat · 2023-12-05
> >
> > **To AC**: Considering the author has acknowledged the drawbacks in clarity and related works, I suggest the author provides a more thorough revision and submits it for a future venue. I would maintain my original score.

---

### Official Review · Reviewer_vkpj · 2023-11-01

**Soundness:** 2 fair
**Presentation:** 1 poor
**Contribution:** 2 fair
**Rating:** 3
**Confidence:** 3

**Summary:**

The paper proposes a method to improve multi-source few-shot domain adaptation in large vision-language models. This new method uses a structured prompt with domain and context prompts to narrow the gap between specific domain and image features, enhancing the model's ability to distinguish between different domains. Experimental results on DomainNet and OfficeHome datasets show performance improvements against the state-of-the-art.

**Strengths:**

The author conducted adequate evaluation to support the proposed approach.

**Weaknesses:**

The paper is challenging to follow, and many of the concepts and motivations are either incorrect or unclear.

**Several claims that are unclear and unsupported:**

1.	“Using learnable prompts increases the risk of overfitting on training samples, which reduces the ability of domain prompt models to extract common semantic features. “ . This is not justified and likely wrong. If training one prompt for a domain do not have the issue, then training multiple prompts respectively for multiple domain should not have the issue as well.

2.	“Large-scale pre-trained vision-language models lack domain awareness during training, which hinders the acquisition of domain-specific tendencies necessary for effective domain adaptation tasks. “. The general training of a model should not influence its adaptation ability.

3.	“These models often misinterpret source domain-specific semantic features as belonging to the target domain, resulting in suboptimal performance for multi-source domain adaptation through prompt learning.” I think prompt tuning is proposed to address the problem that a model use source domain knowledge to complete tasks in target domain.

4.	“We train the prompt of the domain with this mixed-up feature.”. what do you mean “the prompt”? what is "the prompt" referring to?

**Unclear optimization objectives:**

It is unclear what the training objectives of [d] and [v] are. Both t^s and t^k are involved in equations 6 and 7, but it is not clear which prompt is optimized to minimize equation 7.

**Unclear annotation:**

Figure 3: [v] has been used for interpreting context prompts and should not be used to represent domain features. Furthermore, in equation 8, both f_t and f_s are used to represent features from the target and source domains, and the author should maintain consistency in their terminology.

I may not fully understand the paper since the current version is very difficult to follow and understand. I recommend that the paper undergo careful revision before resubmitting it for peer review.

**Questions:**

No additional questions.

---

> ### Author Response · Authors · 2023-11-22
>
> Thank you for your valuable feedback. Your suggestions have significantly contributed to the improvement of my manuscript. I acknowledge that there are several instances in the paper where the expressions are unclear. In this response, I aim to provide concise clarifications to the issues you raised, hoping this will enhance your understanding of my work and the modifications made.
>
> Q1.response:
> Thank you for your valuable insights. We intended to convey that methods similar to CoOp are susceptible to overfitting, posing challenges in training prompt generation models. This concern has been raised in subsequent works. However, due to clarity issues in our expression, the intended message may not have been effectively communicated. We acknowledge this discrepancy and commit to promptly addressing and clarifying this point in our manuscript.
>
> Q2.response:
> In our manuscript, we stated, "Large-scale pre-trained vision-language models lack domain awareness during training, which hinders the acquisition of domain-specific tendencies necessary for effective domain adaptation tasks." I recognize that this statement was unclear and may have led to a misinterpretation. Our intention was to emphasize that in multi-source domain adaptation tasks, where the domain of each image is known, we advocate for the inclusion of domain-specific information in the learning process of large vision-language models. In contrast to approaches that do not consider domain information, such as coop, we conducted a comparative analysis. Our results demonstrate that incorporating domain-aware cues in prompt learning yields improved performance in the context of MFDA.
>
> Q3.response:
> Regarding the mentioned concern, our intention is to emphasize that, given the limited availability of target domain samples, we seek to leverage source domain data to augment and enhance the learning process for the target domain. This strategic use of source domain information aims to bolster our understanding and proficiency in the target domain despite the scarcity of samples.
>
> Q4.response:
> In addition to the aforementioned clarification, I would like to provide further elucidation regarding the statement, "We train the prompt of the domain with this mixed-up feature," particularly concerning the term "prompt." In our context, the term refers to learnable cues, comprising eight 1024-dimensional vectors. To enhance reader comprehension, we have augmented the manuscript with additional descriptions pertaining to these prompts.
>
> Finally, strict modifications have been made to the proposed format and writing errors。
>
> I hope this clarification addresses your concerns regarding the intended meaning of the mentioned statement. I have made corresponding adjustments in the manuscript to enhance the clarity of this aspect.

---

> > ### Comment · Reviewer_vkpj · 2023-12-05
> >
> > The authors' response does not address any my concerns regarding the submission. Besides, although they claim in their response to have made modifications to the paper, no actual revision has been made. I feel the response is solely generated by a LLM. I'd maintain my original score.

---

### Meta-Review · Area_Chair_NixB · 2023-12-08

**Metareview:**

This paper tackles multi-source few-shot domain adaptation (MFDA) in large vision-language models by proposing a method involving domain prompts and context prompts. While the problem addressed is significant, the paper has several weaknesses. The presentation is challenging to follow, with unclear concepts, inconsistent notation, and confusing claims. Novelty is questionable, resembling a previous method (DAPL) extended to multi-source scenarios. The description of the MFDA setting is confusing, and there's a lack of clarity in the optimization objectives. The writing requires substantial improvement, addressing inconsistencies, clarifying notations, and avoiding text overlap with related works. Additionally, the paper lacks comprehensive comparisons with existing approaches, more ablation studies, and clear explanations for various aspects. A careful revision and restructuring are recommended before resubmission to enhance the paper's clarity and overall contribution.

After rebuttal, none of the reviewers was satisfied with the response. AC recommends reject.

**Justification For Why Not Higher Score:**

This is a paper with many theoretical and experimental flaws. After rebuttal, all the reviewers said that the authors haven't addressed their concerns.

**Justification For Why Not Lower Score:**

N/A

---

### Decision · Program_Chairs · 2024-01-16

Reject